# ACapMed: Automatic Captioning for Medical Imaging

Djamila Romaissa Beddiar [1,*] , Mourad Oussalah [1,2,*] , Tapio Seppänen [1] and Rachid Jennane [3]

1 Center for Machine Vision and Signal Analysis, University of Oulu, FI-90014 Oulu, Finland
2 Faculty of Medicine, University of Oulu, FI-90014 Oulu, Finland
3 IDP Laboratory—UMR CNRS 7013, University of Orleans, CEDEX 2, 45067 Orleans, France
* Correspondence: djamila.beddiar@oulu.fi (D.R.B.); mourad.oussalah@oulu.fi (M.O.)

**Abstract:** Medical image captioning is a very challenging task that has been rarely addressed in the literature on natural image captioning. Some existing image captioning techniques exploit objects present in the image next to the visual features while generating descriptions. However, this is not possible for medical image captioning when one requires following clinician-like explanations in image content descriptions. Inspired by the preceding, this paper proposes using medical concepts associated with images, in accordance with their visual features, to generate new captions. Our end-to-end trainable network is composed of a semantic feature encoder based on a multi-label classifier to identify medical concepts related to images, a visual feature encoder, and an LSTM model for text generation. Beam search is employed to ensure the best selection of the *next word* for a given sequence of words based on the merged features of the medical image. We evaluated our proposal on the ImageCLEF medical captioning dataset, and the results demonstrate the effectiveness and efficiency of the developed approach.

**Keywords:** image captioning; medical images; report generation; multi-label classification; LSTM; VGG-16

## 1. Introduction

Since the early ages, images have been acknowledged as ideal media to convey ideas, share opinions, and explain complex concepts through their visual representations and illustrations. In this respect, images act as storytellers that promote and communicate science. From a cognitive perspective, this is rooted back to the functioning of the human brain, which processes visual content faster than textual content as images grab the user's attention much faster than words. With increasing progress being made in digital health technologies, hospitals produce large amounts of medical images from different modalities. Medical images can be used for fast diagnoses and screening of many diseases. Indeed, they hold important data about different pathologies and could be employed to detect abnormalities [1]. Manually extracting information from images or describing their content can be tedious and time-consuming [1], and require the involvement of experts due to possible complexities and multiple interpretations that can be assigned to a single medical image. Therefore, dealing with all images produced by the hospital(s) in a timely manner is challenging and costly. This impacts the ability to produce accurate reports within the required deadline and substantially increases the workloads of the personnel [2]. One solution is automatic captioning of images, which consists of describing, automatically, and in natural language, the visual content of images [3]. Automatic image captioning combines skills from two main fields: computer vision for image processing and natural language processing for text generation. In general, image captioning is employed for diverse applications whenever a textual description is required from visual content, such as automatic image annotation and labeling, video transcription, security detection, and medical image interpretation. The latter plays an important role in computer-aided diagnosis systems, decision-making, and disease treatments by releasing workflows and assisting professionals in their daily routines.

However, medical image captioning is not a trivial process. Automatically generated reports are prone to errors and low-quality textual descriptions (among other issues) [4]. This is due to the nature of medical images, which are not as simple as natural images, as illustrated in Figure 1. Moreover, there are high expectations from these reports to accommodate clinical standards, which require extensive expertise [5]. In general, such reports should follow specific templates, include medical terms, and highlight clinically important information by giving visual evidence rather than describing objects in the image [6]. Notably, classical captioning models struggle to generate accurate descriptions for medical images and still need improvements to be clinically acceptable [3]. Therefore, the implementation of new approaches tailored to medical image captioning is a field of great interest in artificial intelligence. This may help in the fast exploitation of medical content, delivering faster and more accurate interpretations of findings, and providing valuable assistance to doctors by alleviating their workloads and expediting clinical workflows [3].

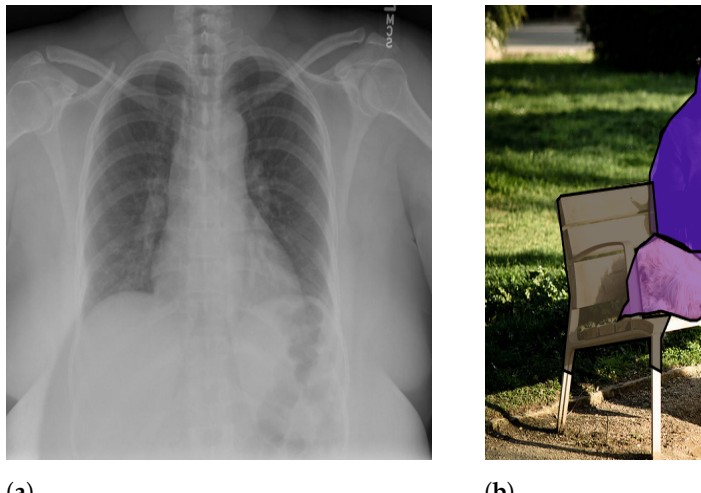

(**a**)        (**b**)

**Figure 1.** Comparison between a medical image and a natural image. (**a**) Represents a medical image captioned as 'Heart size normal, Lungs are clear' and its associated medical concepts are the PA and lateral chest. (**b**) Represents a natural image, captioned as 'man reading a magazine on a park bench, with his dog'. Objects detected are: person, bench, dog, and book. Anyone can generate a natural caption but only experts are able to comprehend the content of medical images and, therefore, generate adequate captions. Moreover, concepts related to medical images are given by experts while objects in the natural image are intuitive.

Although major efforts have been made to enhance the quality of natural image captioning (boosted by commercial and security-related applications), few advances have been made in the medical image captioning field, where accuracy has barely exceeded 40% in several benchmarking competitions [1]. In this regard, in the current paper, we suggest extracting medical concepts from the image content and employing them with the visual features to generate a new caption. The motivation behind this work comes from the use of object detection algorithms in image captioning systems, such as in [7,8]. First, a convolutional neural network (CNN), acting as a multi-label classifier (MLC), was trained on image samples to detect medical concepts. Then, visual features were extracted from images using a pre-trained neural network. Moreover, semantic features were constructed from the MLC outputs, which were then pre-processed and tokenized, and then their embeddings were calculated. Next, the same pre-processing type was applied to the image captions where the vocabulary was constructed from all words of the identified medical concepts and captions. Finally, the visual features and semantic features were merged and fed to a long short-term memory network (LSTM) to generate new caption(s) by relying on the original caption of the image. Moreover, beam search was employed with the LSTM to

select the best outputs while predicting the *next word* from the previous sequence and visual information. The proposed technique was evaluated on a publicly available medical dataset. The results demonstrate the feasibility and the technical soundness of our method. Overall, our key contributions are summarized as follows:

- A new approach to generate *vocabulary* for text generation and encoding was constructed from medical concepts and image captions.
- A multi-label classification MLC model based on VGG-16 is proposed to detect medical concepts from images.
- A pre-trained VGG-16 network was employed to extract visual features from medical images.
- An end-to-end deep learning-based network was employed for text generation, fusing visual and semantic features extracted from images as well as their associated medical concepts. The model merges different networks: MLC for semantic feature encoding, VGG-16 for visual feature extraction, caption embedding, and an LSTM network for caption generation.
- A beam search was employed alongside LSTM to accurately select the best words among the list of predicted words for caption construction.
- An ablation study was conducted to investigate the impacts of each component of the proposed model in the overall performance.

The rest of this paper is organized as follows. First, we introduce the context of our work and briefly summarize state-of-the-art models on medical image captioning in Section 2. Section 3 outlines our approach, which used datasets and evaluation metrics. Then, we discuss the experimental results of our approach in Section 4. Finally, we discuss the outcomes of our approach, conclude the paper, and provide future directions for medical image captioning in Section 5.

## 2. Background

Image captioning is of great interest in many application domains, such as automated vehicles, image retrieval, and healthcare. Indeed, mapping the visual content of an image into text, automatically, without the need for human involvement, frees up the workloads on many annotation systems [9]. In the literature, many research studies have focused on natural image captioning and have not considered medical images since describing the content of a natural image is much easier and less demanding. Indeed, actors, objects, and their relationships can be described using simple statements, which form the *captions*. However, interpreting the findings in a medical image is a challenging task and requires specific expertise [3] because images are obscure and illustrate organs or tissues that only professional practitioners could identify. In addition, relationships between objects in the image are not really what we look for, but clinical findings or abnormalities are more relevant. To date, only a few works have focused on medical image captioning by adapting existing natural image captioning to deal with medical images. In Ayesha et al. [10], the authors categorized image captioning models into three main categories, which can further be applied to medical image captioning approaches. We compare state-of-the-art techniques on image captioning in Table 1 and summarize them as follows:

- *Template-based techniques*: Exploit objects or attributes detected from images and use grammar rules and constraints to generate captions [10]. Templates are filled out with specific text that better enhance the distinction between normal and abnormal findings. The generated captions are often small, grammatically correct, and hard-coded, constraining the variety and flexibility of the outputs [3]. For instance, Onita et al. [11] used kernel ridge regression combined with classification to map a given image to words, describing the content from a dictionary; the authors investigated how text and

the feature extraction model could influence the captioning performance. They evaluated their proposal on the PadChest dataset (https://bimcv.cipf.es/bimcv-projects/padchest/, accessed on 2 October 2022).

- *Retrieval-based techniques*: Are based on the retrieval of images visually similar to the input image from a large dataset, and use their captions to construct a new caption for the input image [10]. Retrieved captions are either combined to create a completely new caption or the most similar caption is employed as a substitute for the new caption. For instance, Xuwen Wang [12] proposed using the Lucene Image Retrieval (LIRE) system to extract similar images of given medical images based on their underlying concepts, which were detected using a multi-label classifier and a topic modeling method. Then, semantic annotated concepts were combined with the body parts of images to cluster them into different groups. The proposed technique could be extended to retrieve the most relevant captions of the inputted image from captions of images of its class.

- *Deep learning-based techniques*: Rely on the end-to-end trainable networks to extract automatic features from images and map them into meaningful text [3]. Since deep learning-based models performed very well for many other domains, this category is the most investigated in image captioning as well. These techniques include encoder–decoder architectures, fully connected networks, and CNNs [10]. The existing methods in the literature are basically inspired by the *show-and-tell* model [13], an encoder–decoder model for image captioning. The proposed technique is based on a visual feature extractor (the encoder), which is usually a CNN network, and a text generator, which is an RNN network (the decoder). This model is further improved to deal with medical images and to focus on important parts of the image (parts where the most significant features are obtained) using attention mechanisms. For instance, the authors of Zeng et al. [1] proposed detecting lesion areas, extracting automatic visual features from them, diagnosing pathological information, and reporting the findings from medical images using an encoder–decoder architecture. Similarly, Yin et al. [14] employed a deep CNN-based multi-label classifier as the encoder and a hierarchical RNN as the decoder. First, the model detects abnormalities in medical images, which are then used to generate long medical annotations based on an attention mechanism. Likewise, Beddiar et al. [15] exploited the show–attend–tell model [16] (extension of the show-and-tell model, using attention mechanisms) and changed the decoder with a GRU network for medical image captioning. In contrast to encoder–decoder architectures, visual and semantic features were fused and exploited to generate captions in merged models. In general, merged models employ CNN for visual feature extraction and an RNN for textual features. Then, textual and visual features are merged to assign a new caption to a given medical image. Obviously, they constitute a variant of encoder–decoder architecture with the fusion of textual information. For instance, Wang et al. [17] employed a joint representation of image–text pairs calculated using a variational autoencoder model for medical report generation. They used a topic model theory to model semantic topics of images that were exploited, in addition to deep fuzzy logic rules designed based on diagnosis logic, to summarize and interpret abnormalities in medical images. Likewise, Al Duhayyim et al. [18] proposed using encoding and decoding architectures for the generation of effective captions where SSA-based hyperparameter optimizers in both parts were used to attain effective results.

Some works combined techniques of the previously enumerated categories to accurately generate descriptions, which were more relevant and focused on. For instance, Xie et al. [19] proposed a topic-guided attention mechanism to generate descriptions for abnormal visual observations by giving evidence on locations of the abnormalities. Moreover,

Li et al. [20] proposed abnormality graph learning, allowing them to detect abnormalities and use them to retrieve text templates, which were further adapted, enriched, and corrected using a paraphraser. In Li et al. [6], the authors proposed a hybrid retrieval-generation reinforced agent that decided whether to retrieve a template sentence or generate a new one for each constructed topic related to medical images. Similarly, in Zhao et al. [21], the authors proposed a model that incorporated a generation-based approach and a retrieval-based method. It employed visual features and retrieved similar textual features to generate the final captions.

**Table 1.** Comparison of some state-of-the-art methods on image captioning.

| Method | Input | Output | Dataset |
|---|---|---|---|
| Template-based method: kernel ridge regression and classification [11] | chest X-ray images | description | PADChest dataset |
| Retrieval-based method: Lucene Image Retrieval (LIRE) + multi-label classification and topic modeling [12] | medical images | description | ImageCLEF + ROCO datasets |
| Deep-learning-based method: show and tell encoder–decoder model [13] | natural images | captions | Pascal VOC 2008, Flickr8k, Flickr30k, MSCOCO |
| Deep-learning-based method: encoder–decoder model [1] | medical images | diagnosis report | IU X-ray + own created dataset |
| Deep-learning-based method: encoder–decoder model with CNN-based multi-label classifier and RNN as decoder [14] | chest X-ray images | medical report | IU X-ray dataset |
| Deep-learning-based method: encoder–decoder with attention, show, attend and tell model [16] | natural images | captions | Flickr8k, Flickr30k and MS COCO |
| Deep-learning-based method: encoder–decoder with attention, decoder with GRU [15] | medical images | descriptions | ImageCLEF 2021 |
| Deep-learning-based method: merge model with a variational auto-encoder [17] | chest X-ray images | medical report | IU X-ray dataset |
| Deep-learning-based method: encoder–decoder with SSA-based hyperparameter optimizer [18] | natural images | captions | MSCOCO and Flick8K |
| Hybrid method: topic-guided attention mechanism [19] | lateral and frontal chest X-ray images | description | IU X-ray dataset |
| Hybrid method: abnormality graph learning with retrieval technique [20] | lateral and frontal chest X-ray images | medical report | IU X-ray dataset |
| Hybrid method: method based on a Retrieval-Generation Reinforced Agent [6] | lateral and frontal chest X-ray images | medical report | IU X-ray + ChexPert dataset |
| Hybrid method: retrieval-based method combined with generation-based method [21] | natural images | captions | MSCOCO |

### 3. Materials and Methods

In this paper, we present a captioning technique that combines visual features extracted from images and their associated medical concepts for medical image captioning (we were motivated by the existing object detection-based methods for natural image captioning). Indeed, applying object detection methods on medical images does not provide satisfactory results and, therefore, could not be exploited for automatic captioning. However, an alternative to that is extracting medical concepts associated with medical images using some pre-trained models. Outputs of such systems could help to improve the reliability of existing medical image captioning systems. For this reason, we propose merging visual features and semantic features in order to generate new captions. Visual features are computed from images using some pre-trained networks. Semantic features are computed from medical concepts detected for medical images. CNN networks are used for feature extraction for both visual and semantic cases whereas a multi-label classifier is implemented to detect the concepts. Finally, an LSTM network is implemented for language generation where beam search is also employed for a better selection of words predicted to construct the caption. In the following subsections, we present the different steps of our proposal, which are illustrated in Figure 2.

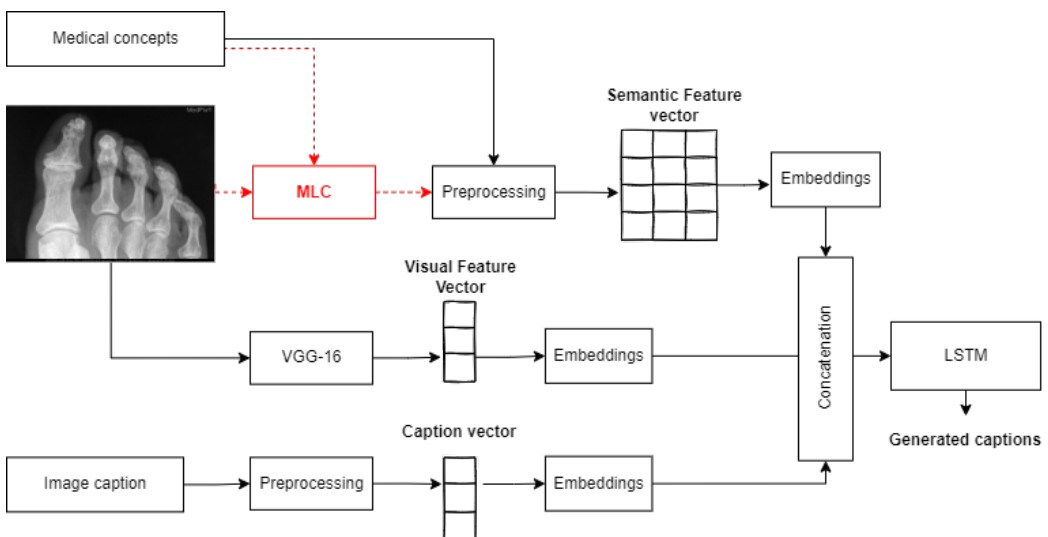

**Figure 2.** Overall scheme of our proposal. Visual features were extracted from medical images to construct the visual feature vector and train the multi-label classifier (MLC). The semantic features were computed from encoding the outputs of the MLC, combined with visual features and caption vector, and input to the LSTM for text generation. The diagram without the red box shows the training process; it shows the testing process when the red box is included. The medical concepts were not inputted directly for pre-processing but were passed through the MLC.

### 3.1. Visual Feature Encoding

The first step of our proposal is visual feature extraction; we employed a pre-trained CNN model. We used the VGG-16 model since it is small, has been trained on the large ImageNet dataset, and performs very well on several other classification tasks. The model is composed of 16 layers, to which we input the medical images and generated a feature vector of size 4096 after removing the last classification layer, as illustrated in Figure 3. The features extracted from this layer were learned by the model while trying to predict the image class and distinguish the visual content of images. Images were pre-processed before fitting them to the VGG-16 model. They were normalized and resized to fit in the encoder and augmented with some traditional image augmentation techniques.

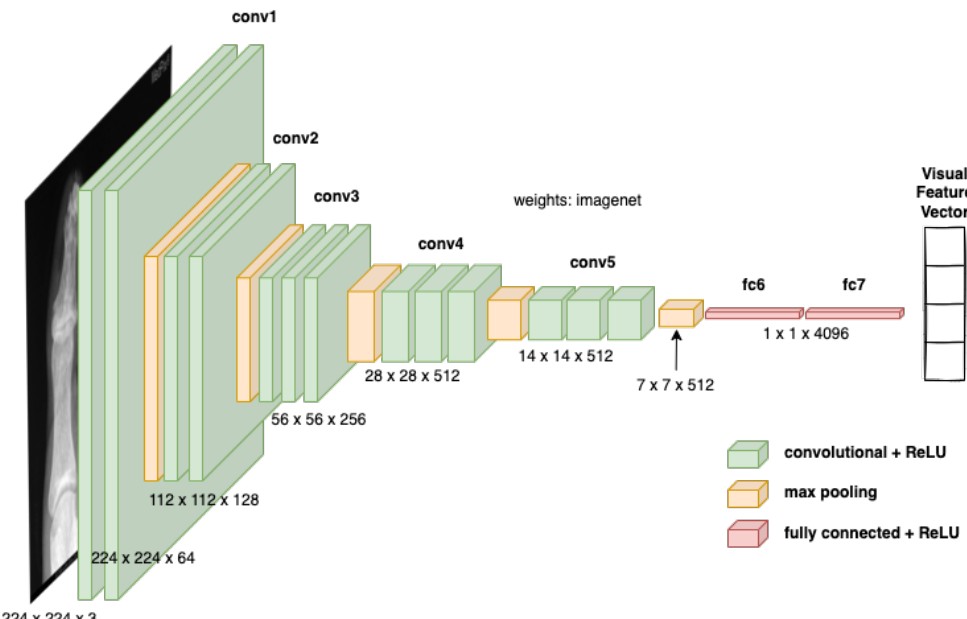

**Figure 3.** The model for visual feature extraction. The model inputs were medical images and the outputs were the visual feature vectors (size: 4096).

### 3.2. Semantic Feature Encoding

#### 3.2.1. Text Pre-Processing

We pre-processed the captions to clean the text and keep only significant words. Specifically, we tokenized each caption, converted all characters to lowercase, removed stop-words, filtered out the punctuation, and calculated the stems of the identified words. Two other words were added to each caption to identify the beginning (*<start>*) and the end (*<end>*) of the sentence. This pre-processing process was performed using the NLTK package (https://www.nltk.org, accessed on 2 October 2022). The maximum length of one caption for the whole training set was 50 words, which was used to pad the remaining captions at the end. Further, embeddings were calculated from these captions to capture the semantic meaning of each sentence. Finally, each image was represented with a vector of size 50, encoding the corresponding caption.

As previously noted, we used both visual features and semantic features as joint features for the process of caption generation. So, semantic features were obtained from the processing of the medical concepts associated with the images. We processed them in the same manner as we did for the captions. Medical concepts are provided as concepts unique identifiers (CUIs), e.g., C1306645 refers to 'Plain X-ray'. The first step was to substitute each CUI with its corresponding text (UMLS—unified medical language system), which is possible through the National Library of Medicine (https://www.nlm.nih.gov/research/umls/index.html, accessed on 2 October 2022). In general, the UMLS constitutes different words; we tokenized, converted to lowercase, stemmed, and removed the stop words among them. Consequently, each image is represented with a vector (of a shape of 10x9), where ten concepts were considered and each concept was encoded with a vector of size 9 (composed of 9 words). The maximum number of CUIs per image was found to be 10; after tokenization, the maximum length of one CUI was 9 words for each concept of the training set. If the concept had fewer words, we padded the sequences at the end. Similarly, if the image had fewer associated CUIs, we repeated the vector of the last CUI for the remaining positions (to reach 10 CUIs, i.e., 10 vectors). Embeddings were calculated as well from medical concepts and constituted the semantic features.

### 3.2.2. Vocabulary Construction

Now that tokens were generated from captions and concepts, we constructed the vocabulary. A dictionary of sorted unique words was constructed so that a numerical value was assigned to each word based on its order in the set.

### 3.2.3. Multi-Label Classification for Medical Concept Detection

We propose using a multi-label classifier to detect and recognize concepts from medical images. For that, we trained a CNN network that was inspired by the VGG-16 network but on a smaller scale. Inputs of the model were the medical images and outputs were multiple classes, which linked to the identified concepts from visual features. We picked the top ten recognized concepts to represent each image. The predicted concepts were further pre-processed and constituted the basic units of our semantic feature encoding process. Figure 4 represents the architecture of our multi-label classifier.

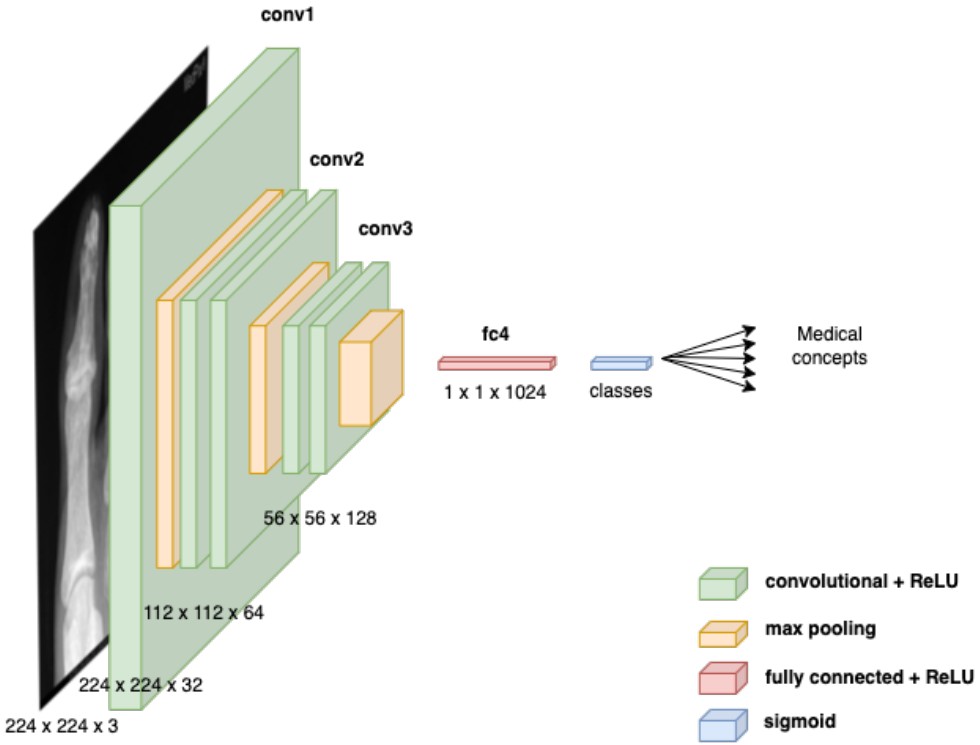

**Figure 4.** Architecture of our proposed multi-label classifier. Image was inputted to a smaller VGG-16 network and multiple medical concepts were outputted.

### 3.3. Captioning Model

The ultimate goal of our model was to generate descriptions from the given medical images and their associated concepts. Our captioning model was composed of a visual feature encoder, a semantic feature encoder, a multi-label classifier acting as a semantic feature encoder, as well as an LSTM for caption generation. Outputs of the encoders were converted to the same size (128), merged, and inputted into the generation model. In other words, the input to our model was $[x_1, x_2, x_3]$ where $x_1$ is the 4096 feature vector of the image, $x_2$ is the (10x9) semantic vector encoding the concepts, and $x_3$ is the semantic vector encoding the caption sequence. The output of the model was $y$, which is the next word that the model should predict using the three inputs as illustrated by Table 2. An example of the generation of the caption 'macroscopic fat containing nodule in the right adrenal gland' is shown, where the sequence was predicted word-by-word using both visual and semantic feature vectors.

**Table 2.** Example of how our model generates the training sequences.

| $x_1$ (Visual Vector) | $x_2$ (Semantic Vector) | $x_3$ (Text Sequence) | $y$ (Word to Predict) |
| --- | --- | --- | --- |
| visual feature | semantic feature | start, | macroscopic |
| visual feature | semantic feature | start, macroscopic, | fat |
| visual feature | semantic feature | start, macroscopic, fat, | containing |
| visual feature | semantic feature | start, macroscopic, fat, containing, | nodule |
| visual feature | semantic feature | start, macroscopic, fat, containing, nodule, | in |
| visual feature | semantic feature | start, macroscopic, fat, containing, nodule, in, | the |
| visual feature | semantic feature | start, macroscopic, fat, containing, nodule, in, the, | right |
| visual feature | semantic feature | start, macroscopic, fat, containing, nodule, in, the, right, | adrenal |
| visual feature | semantic feature | start, macroscopic, fat, containing, nodule, in, the, right, adrenal, | gland |
| visual feature | semantic feature | start, macroscopic, fat, containing, nodule, in, the, right, adrenal, gland, | end |

Our model predicts (at each time step) a word from the vocabulary to be next in the sequence. In general, the word with the highest probability is considered even though it may lead to a caption that is not really suitable for the image since it only looks one step back. Therefore, we investigated the beam search algorithm, which allowed us to consider all possibilities for a generated sequence and progressively choose the best ones by looking at different steps of sequence generation. We explored three different values of the beam width $k$, which specified the number of best possibilities to be sent to the next step. So, $k = 3$, $k = 5$, and $k = 7$ were employed and the process was repeated for each word until the max length of the caption or the '<*end*>' token was reached.

During the training, we input the medical images, their pre-processed concepts, and their corresponding pre-processed captions. An LSTM network was then used as the language generation model since it performs well for sequence prediction problems. LSTM predicts the next word in a sequence based on the previous word, the semantic, and the visual features of each image until it reaches the end of the sequence given by the token <*end*>. Figure 2 illustrates the training process (by removing the red box, because ground truth concepts are used directly from the training set).

During the test phase, the model first predicted the concepts associated with the medical image (using the multi-label classifier), and then carried on with the encoding of visual and semantic features as shown in Figure 2. Finally, it output the predicted captions, word by word. Beam search was used at each step to select better-predicted words for the sequence previously constructed.

### 3.4. Dataset

To evaluate our proposed method, we employed the ImageCLEFmed 2021 dataset [22,23], composed of three subsets for training, validation, and testing. It is composed of 2756 medical images where each image is associated with a set of medical concept unique identifiers (CUIs) and a caption consisting of one or several sentences. CSV Files (mapping images to their corresponding CUIs and captions) are provided with the raw images. Multiple image modalities and body parts are considered in this dataset, making it very challenging and more general. Figure 5 shows two samples of image–text pairs from the ImageCLEFmed 2021 dataset. The UMLs are retrieved to identify each CUI of the image. They are related to the content of the image and could give more insight into its content. For example, in the left sub-figure, CUI 'C0003842' is substituted by 'Arteries' and the CUI 'C0040398' by 'Tomography, Emission-Computed'. Both concepts give more information on the content of the image and could be used to boost the understanding of the image.

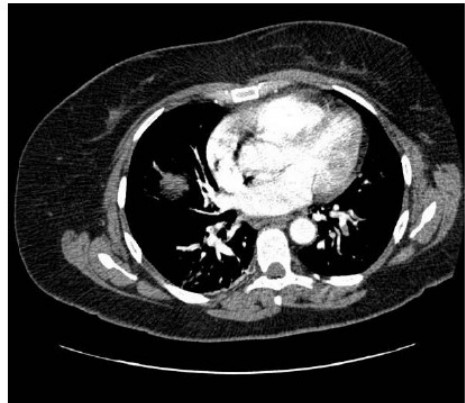
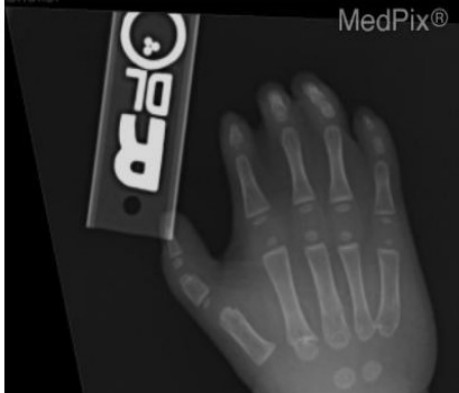

ImageID: synpic37363
Caption: Filling defects in the segmental arteries of the right and left lower upper lobes consistent with pulmonary emboli.
CUIs: C0003842; C0040398
UMLS: Arteries; Tomography, Emission-Computed.

ImageID: synpic27940
Caption: Soft tissue webbing between fingers of the right hand.
CUIs: C0225317; C0221352; C0230370; C1306645.
UMLS: Soft tissue; Syndactyly of fingers; Structure of right hand; Plain x-ray.

**Figure 5.** Radiology image samples from the ImageCLEF dataset; captions describe the images, and their associated CUIs are presented.

### 3.5. Evaluation Metrics

Our model is composed of different models that can be evaluated to investigate the performance of each component. First, we evaluated the accuracies of the MLC, LSTM, and the quality of the generated captions using the bilingual evaluation understudy (BLEU) metric. It allowed us to compute the similarity between the reference caption and the hypothesis proposed by the system. It varied between 0 and 1; 0 referred to no overlap between translation and reference (low quality), and 1 referred to the perfect overlap between them (high quality).

For the MLC model, we calculated the *F_measure* as follows:

$$F\_measure = 2 \cdot \frac{Precision \cdot Recall}{Precision + Recall} \tag{1}$$

where the recall and the precision are calculated as follows:

$$Recall = \frac{TP}{TP + FN} \tag{2}$$

$$Precision = \frac{TP}{TP + FP} \tag{3}$$

In general, *TP*, *FN*, *TN*, and *FP* refer to true positive, false negative, true negative, and false positive, respectively.

The accuracy, which provides insight into the ability of the model to correctly predict the associated concept, is given by:

$$Accuracy = \frac{\text{Correct Predictions}}{\text{Total Predictions}} = \frac{TP + TN}{TP + FN + TN + FP} \tag{4}$$

Finally, the *BLEU* metric is calculated as follows:

$$BLEU = BP \cdot \exp\left(\sum_{n=1}^{N} w_n \cdot \log p_n\right) \tag{5}$$

where $N$ refers to the number of *n_grams*, and by default, it is 4, $W_n$ indicates the weight of each modified precision and it is set by default to $1/4$, while $P_n$ refers to the modified precision, which is calculated as follows:

$$p_n = \frac{\displaystyle\sum_{H \in \{hypotheses\}} \sum_{n\_gram \in H} Count_{clip}(n\_gram)}{\displaystyle\sum_{C' \in \{hypotheses\}} \sum_{n\_gram' \in C'} Count(n\_gram')} \tag{6}$$

where $Count_{clip}$ calculates the count of clipped *n_gram* of the hypotheses and *Count* gives the number of hypothesis *n_grams*.

Moreover, *BP* or the brevity penalty is used to select the caption, which is more similar to the reference caption, in length, word choice, and order. It is given by:

$$BP = \begin{cases} 1 & h > r \\ \exp^{(1-r/h)} & h \leqslant r \end{cases} \tag{7}$$

where $r$ and $h$ refer to the number of words in the reference caption and the hypothesis caption.

## 4. Results

We investigated the accuracy of the LSTM model for generating captions. The model was trained for 100 epochs; the accuracy is reported on the test set. Figure 6 illustrates the training accuracy and loss where we can see that the model learned to generate the next words for each caption sequence over time. Indeed, the accuracy was improved over epochs and the loss decreased.

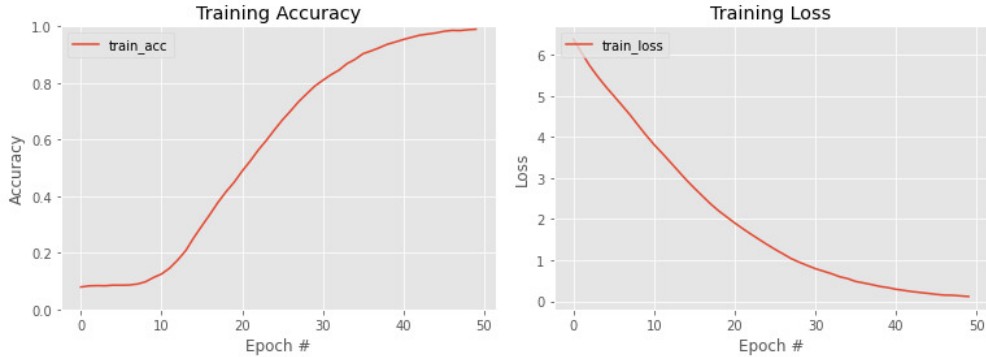

**Figure 6.** Performance of the LSTM language generation model. Left side: training accuracy; right side: training loss. Epoch # refers to the number of epochs.

Moreover, the BLEU score was calculated for the set of generated captions as demonstrated in Section 3.5. We reached a BLEU score of 42.28% for the captioning using beam search with index = 5. We illustrate the results of different experiments (regarding how each

word in the caption is generated) using argmax and beam search with index = 3, 5, and 7. Better results were obtained with beam search and beam index = 5 while other indices gave similar values. Results are shown in Table 3, and are also compared to other state-of-the-art techniques on the same dataset. We can see that the model proposed by [24], a winner of the ImageCLEF challenge in 2021, gave the best BLEU result with a value of 51.00%. In Castro et al. [24], the authors introduced a multi-label classification-based method, where each word in the caption was considered to be a class of the image. The authors then trained their model to predict high-ranking words (i.e., classes of the image) based on a statistical method and then used them to construct the caption. Our results outperformed those of [25], which employed a pattern-based captioning method by combining the medical concepts identified for the image and achieved a BLEU score of 25.70%. Captions were then created based on the characteristics of captions in the training and validation sets and depended marginally on the outcomes of the model, which used concept identification. Similarly, we outperformed the results of [26], where an encoder–decoder model was presented with images as input and an attention mechanism between the encoder and the decoder. The model was inspired by the famous 'Show, Attend, and Tell' [16] model, where the image encoder was changed into a ResNet-101 network. However, this model does not employ medical concepts for the captioning process.

**Table 3.** Comparison of our results to some state-of-the-art results.

| Method | BLEU |
| --- | --- |
| ImageSem [25] | 25.70% |
| IALab_PUC [24] | **51.00**% |
| Kdelab [26] | 36.20% |
| ours: MLC + LSTM + Argmax | 41.09% |
| ours: MLC + LSTM + Beam Search 3 | 42.15% |
| ours: MLC + LSTM + Beam Search 5 | 42.28% |
| ours: MLC + LSTM + Beam Search 7 | 41.16% |

In order to further evaluate the performance of our model, we show some examples of images from the validation set and both their ground-truth and generated captions in Figures 7–9. We can see that our model was able to accurately generate captions in both upper sub-figures and the lower left sub-figure of Figure 7, where all words were correctly predicted (i.e., the caption 'left upper lobe mass' for the example was correctly generated by our model). Even the order of the words in the caption was correct. However, in the lower right sub-figure of Figure 7, although the wording appears to be relevant and correctly generated, their ordering was wrong and, hence, the caption was incorrect as well. Moreover, the model generated some wrong words (in red) as in Figure 8 (i.e., words such as 'heterogeneous' and 'crossing the' in the lower left sub-figure were wrongly inserted in the caption, affecting the quality and the correctness of the interpretation). The model missed some words (in red), such as in the upper right sub-figure (i.e., 'material' and 'Note the large amount of subcutaneous fat circumferentially compatible' were missing). The model was trained to generate one sentence for each image and that is why it was not able to generate a caption composed of more than one sentence, which could be the case for many long medical reports. Moreover, in some cases, the model generated completely wrong captions as in Figure 9. Likewise, it appears that some reports used sentences that were too long, which lowered the quality of the embedding generated by the used encoders, and thereby, led to wrong captions.

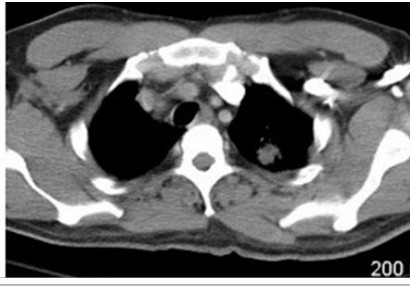
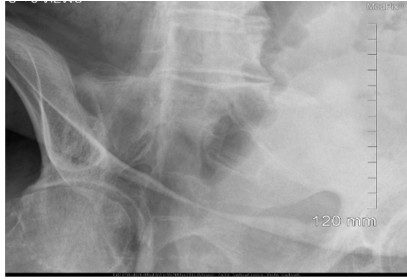

| Ground-truth: Left upper lobe mass.<br><br>Generated: left upper lobe mass | Ground-truth: Fusion of multiple disc spaces Squaring of the vertebral bodies Fusion of SI joints.<br><br>Generated: fusion of multiple disc spaces squaring of the vertebral bodies fusion of si joints |

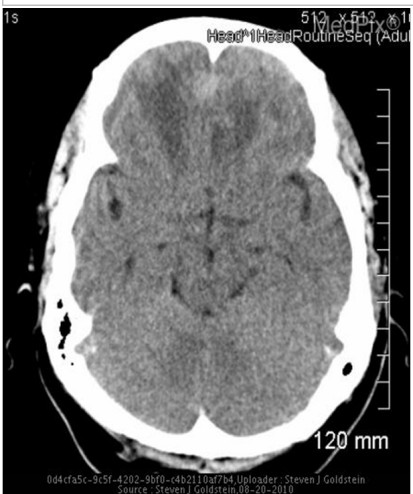
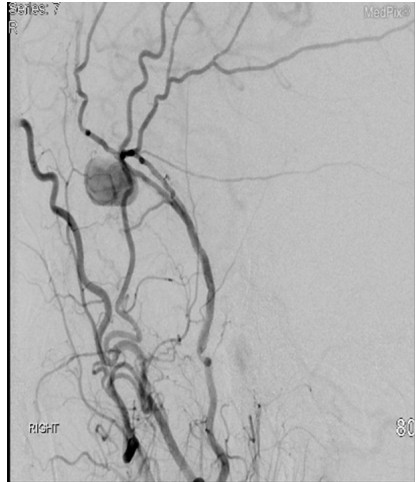

Ground-truth: CT-edema both frontal lobes/ MR—Diffuse thickening of the basal meninges with intense enhancement. Edema base of frontal lobes. Pan sinusitis.

Generated: ct edema both frontal lobes mr diffuse thickening of the basal meninges with enhancement edema base of frontal lobes pan sinusitis

Ground-truth: The catheter angiogram shows a 1.5cm aneurysm arising from the right middle meningeal artery branch of the external carotid artery.

Generated: right catheter of the right branches is arising from the aneurysm branch to the right meningeal artery branch of the carotid artery

**Figure 7.** Some samples of captions generated by our model. Upper sub-figures and the lower left illustrate correctly generated captions while the lower right sub-figure illustrates a partially correct caption (but the order of the words is mixed). We illustrate in red words that were wrongly inserted or missed by the model.

*Ablation Study*

To evaluate the whole performance of our proposed model, it is worth investigating each component separately. This could tell if the combination positively or negatively influences the performance of the overall model.

We first investigated the performance of the proposed multi-label classifier. In other words, we evaluated the performance of the proposed CNN in predicting accurate medical concepts from medical image inputs. The model was trained for 100 epochs and the accuracy was reported on the test set. Figure 10 illustrates the training and validation

loss as well as the accuracy of the proposed model. Accuracy increases over time while loss decreases, which means that our model is able to learn to assign multiple labels to the input images. We obtained an *F_measure* value of 41.80%, which was not the best one compared to some state-of-the-art methods. However, this value was obtained because we considered 10 medical concepts for each image, which was the maximum number of concepts for images of the used dataset. As can be seen in Figure 11, regarding the distribution of CUIs per image, most of the medical images only had two CUIs. In contrast, very few images had either one or more than five CUIs. The results, in terms of *F_measure*, are compared in Table 4. Our results are comparable to the state-of-the-art results on the ImageCLEF 2021 dataset. Again, the best results (50.50%) were obtained by the same team Schuit et al. [27] for caption prediction, where a VGG network was used to encode medical concepts. A K_nearest neighbor algorithm was used with perceptual similarity to select concepts that are most likely closest to the given image. In Wang et al. [25], the authors used a pre-trained network for encoding the medical concepts within a multi-label classification task. The medical concepts were divided into four semantic categories, namely imaging type (IT), anatomic structure (AS), findings (FDs), and others; therefore, images were annotated accordingly. Similarly, Jacutprakart et al. [28] achieved an *F_measure* of 41.20% by using a pre-trained DenseNet-121 model for the multi-label classification process after categorizing the medical concepts into different semantic types. Next, two types of training were performed to classify the semantic type and predict the appropriate concept. In Beddiar et al. [15], the authors employed a multi-label classifier, where features were extracted using MobileNet-V2 and then a GRU network was used for the classification.

**Table 4.** Comparison of our results to some state-of-the-art results.

| Method | F_measure |
|---|---|
| IALab_PUC [24] | **50.50%** |
| ImageSem [25] | 41.90% |
| NLIP-Essex-ITESM [28] | 41.20% |
| Attention-based encoder–decoder [15] | 28.70% |
| Our proposal | 41.80% |

Next, we investigated the performances of the medical concepts features rather than only using visual features. So, we investigated the LSTM accuracy and computed the BLEU score of the generated captions without using medical concepts and compared the results to those of using them. The model illustrated by Figure 12 was used for this study. Table 5 shows that when the model used only visual features, the obtained BLEU score was lower than in the case of incorporating medical concept features. The BLEU score was improved because the medical concepts boosted the semantic representation of the images and, hence, increased the ability of the model to generate new captions. We calculated the BLEU score using the argmax and the beam search with indexes 3, 5, and 7, as we did previously. We can see that the best value was obtained for our model using beam search with index = 3 for the best selection of the predicted word. With index = 5, we were able to obtain a very close BLUE score of 36.23% even though the results of the four methods were relatively close, ranging from 33.63% to 36.28%. However, incorporating medical concepts as complementary data to the images increased the BLEU score from 36.28% to 42.28%.

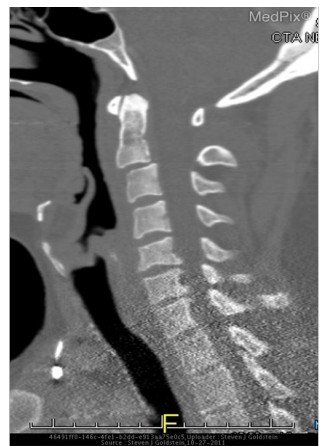
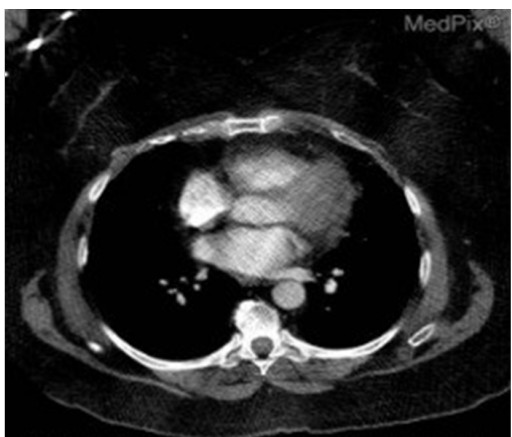

Ground-truth: Fracture dislocation though the C2-C3 disc. Fractures involving the transverse foramenaat C2.

Generated: fracture dislocation at cx at edema and arch of cx

Ground-truth: Axial CT images with intravenous contrast material demonstrate mild smooth thickening of the interatrial septum with diffuse fat attenuation. Note the large amount of subcutaneous fat circumferentially compatible.

Generated: axial ct images with intravenous contrast demonstrate mild smooth thickening of the interatrial septum

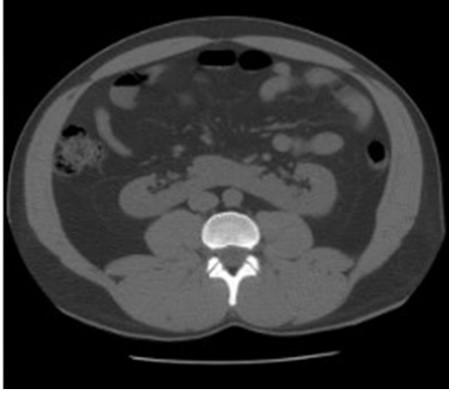
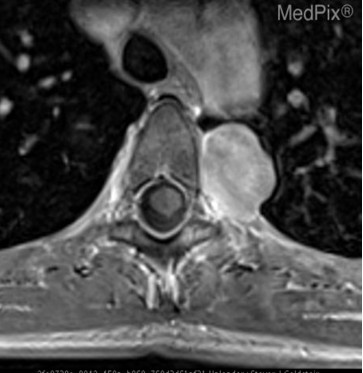

Ground-truth: Fused lower renal poles crossing the midline

Generated: fused heterogenous crossing the poles crossing the midline

Ground-truth: well circumscribed mass adjacent to T3-4. Mass enhances following contrast.

Generated: posterior circumscribed mass adjacent and adjacent cord mass following contrast

**Figure 8.** Some samples of captions partially generated by our model. We illustrate in red some words that were wrongly inserted or missed by the model.

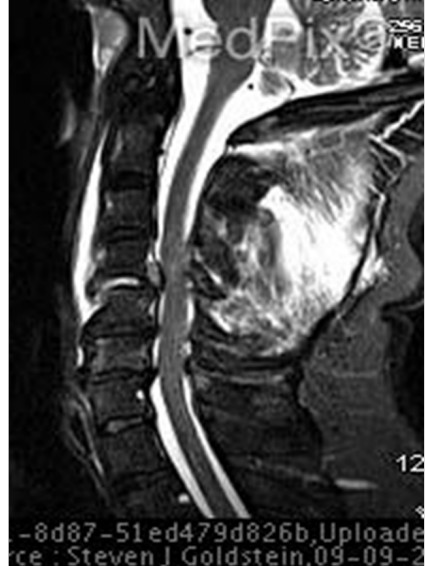
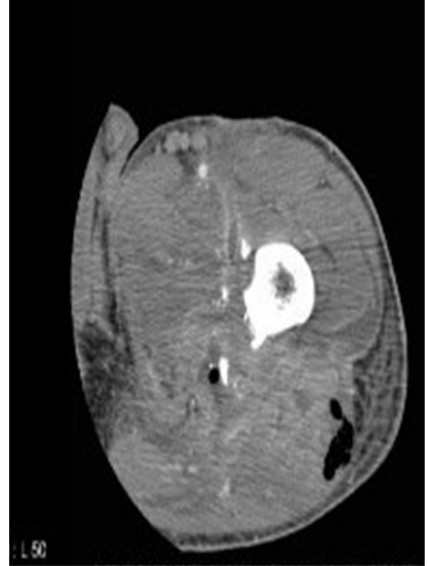

Ground-truth: Fracture through the left C4 lateral mass and laminar arch with unilateral perched C4-5 facets on the right. Herniated and disrupted disk C4-5. Torn intracapsular ligaments and ligament flavum. Prevertebral soft tissue hematoma. Spinal cord compression.

Generated: large lesion with spinal cord at associated spinal cord level

Ground-truth: Axial CT images of the left inguinal region demonstrate an irregular draining fluid collection with enhancing margins extending from the left inguinal region posteriorly to the buttock, with irregular collections of gas extending along the gluteus maximums.

Generated: diffuse thickening of meninges with enhancement

**Figure 9.** Some samples of wrong captions generated by our model. We illustrate in red some words that were wrongly inserted or missed by the model.

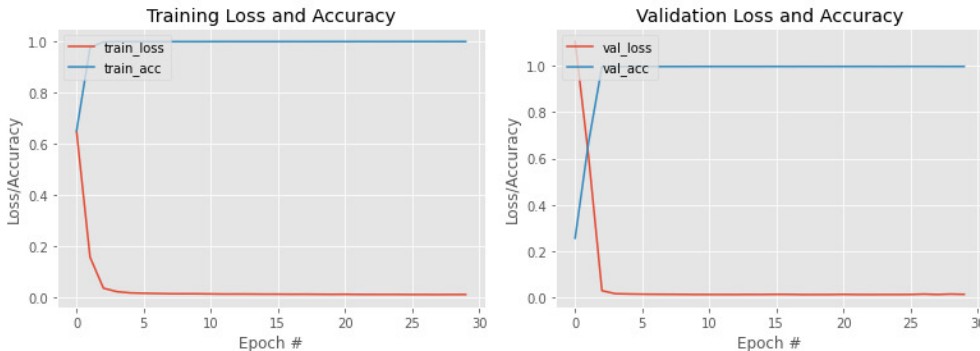

**Figure 10.** Performance of the MLC model. Left side: training loss and accuracy; right side: validation loss and accuracy. Epoch # refers to the number of epochs.

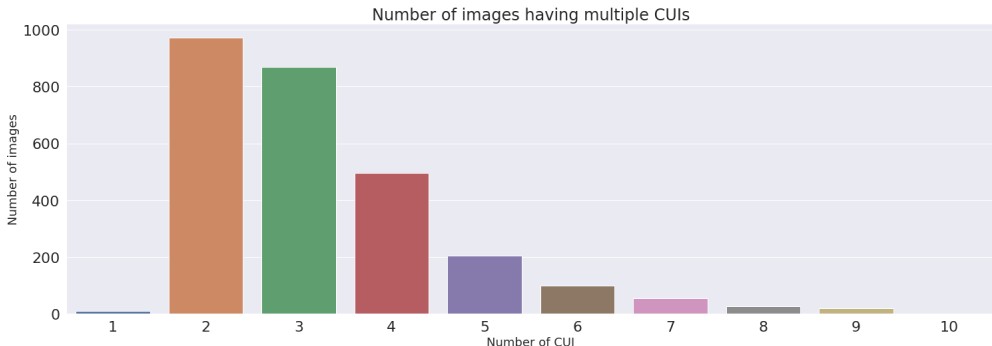

**Figure 11.** Distribution of CUIs per images.

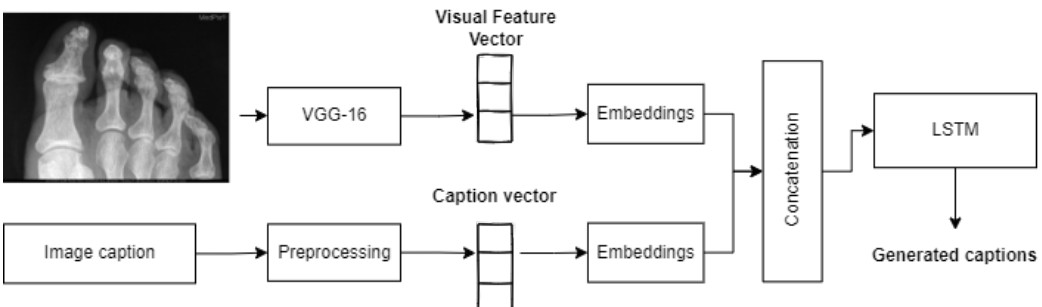

**Figure 12.** Model used for ablation study to investigate the impact of incorporating medical concepts for caption generation.

**Table 5.** Comparison of our results to some state-of-the-art results

| Method | BLEU |
|---|---|
| ours: LSTM + Argmax | 33.69% |
| ours: LSTM + Beam Search 3 | **36.28%** |
| ours: LSTM + Beam Search 5 | 36.23% |
| ours: LSTM + Beam Search 7 | 33.63% |

## 5. Discussion

We proposed an end-to-end model for medical image captioning where medical images and their associated medical concepts were concatenated and fitted to the LSTM model to generate more accurate captions. A visual encoder based on a pre-trained model was used for visual feature extraction and a multi-label classifier was trained to detect medical concepts based on the visual features. Semantic features obtained from medical concepts and image captions were merged with visual features to predict consecutive words of the captions. The main objective of the study was to investigate the impact of incorporating medical concepts related to images to provide more information on the content of the image and, hence, boost the captioning process performance. It was observed that, as expected, the medical concepts offered more information on the image and the captioning accuracy was improved. To achieve this, the medical concepts were pre-processed, encoded, and fitted to the language generator in accordance with the visual features, and the prediction took into account both aspects. However, the model performance was still relatively low and the captioning still needed to be enhanced. Indeed, the model was able to accurately generate short captions but failed in constructing long captions or captions with more than one sentence. Moreover, multi-label classifier performances have a large impact on the performance of the model. In other words, the number of concepts to be predicted or the fixed probability for selecting concepts per image influenced the outcome of the captioning. So, it was essential to find optimized values for how many concepts were predicted for each image. The method used for the selection of the predicted word by observing the next steps

could also influence the captioning accuracy by predicting the best caption rather than only the word with the highest prediction probability. We reviewed in the following some partially or wrongly generated captions and categorized them according to the possible reasons of misinterpretation.

*Error Analysis*

Automatically generating captions from the input image is highly biased by the quality of this image. Indeed, if the visual content of the image is not clear enough, the system will fail in understanding and describing it. We show in Figure 13 some samples of wrong captions where our model either fully or partially failed to generate accurate captions from bad quality image(s). From the left-upper sub-figure, we can see that the image has four sub-images, which would mislead the model while extracting the visual features. From the right-upper sub-figure, we can see that the image has a white background on the right side and the bottom, which would ultimately yield different kinds of features if the same organ is present in another image that is center-cropped, for example. From the left-lower sub-figure, we can see that the image has some writing on the right and some other type of data on the bottom, which influence the feature extraction process. Finally, the right-lower sub-figure shows a foggy image and the features extracted would obviously depend on the visibility of the object in the image that we cannot distinguish easily from this image. We can see that the model was able to identify the objects in the figures but could not create a correct caption, such as in the two right sub-figures. The model succeeded in producing the words 'axial ct demonstrates a rounded mass in the right upper' for the upper sub-figure and 'delayed image, demonstrates tracer activity' for the lower sub-figure.

Similarly, the reliability of the generated captions depends on the words constituting the original captions and their frequency in the dataset. For instance, if the word reoccurs in the dataset, (i.e., present in many captions from the dataset), the model could easily predict that word, but it would likely fail if the word is less frequent or present only in one caption. We illustrate in Figure 14 some samples of the generated captions where the original captions constitute rare words. Even though the system was able to predict that the left sub-figures are a radiograph of the chest and a knee, respectively, it was not able to predict efficient captions because of the words 'parahilar', 'perobronchial', 'tricompartment', 'patellofemoral', and 'cartilage', which are rare in the dataset. Similarly, the model failed in generating good captions for the right sub-figures because of rare words, such as 'cartilage', 'trochlear', 'groove', and 'hyperintensity', or unknown words, such as 'extramuscular' and 'forefoot', which do not exist in the training captions. Roughly speaking, the model can predict words that are recurrent and thereby have a higher likelihood to construct the caption rather than a word that does not exist or rarely appears in the training dataset because our model is not well trained on the embeddings.

We show in Figure 15 recurring and rare words from the training captions. Words, such as 'emission', 'tomography', and 'computed' are the most recurrent in the training dataset; the model could smoothly identify their embeddings and predict them because it was well trained on them. However, words, such as 'tarsal', 'bolus', and 'micronodule', were crossed by the model only once and, thereby, were not predicted accordingly.

Furthermore, some captions include numerical information or punctuation that are semantically important. However, by removing the punctuation, stop words, and some of the characters during pre-processing, we may lose some important information, which in turn affects the meaning and the generation of the new caption. Figure 16 demonstrates samples of original captions containing specific number tokens that, if removed, influence the generation of the new captions. For instance, in the left sub-figure, 'day 10' has an important clinical role in the diagnosis since it presents the historical data of the patient. Similarly, in the right sub-figure, (<4 mm) is important information about the size that influences the diagnosis. In both figures, the wrongly generated captions may be the result of removing the prime information or the length of the original captions, which are composed of more than one sentence.

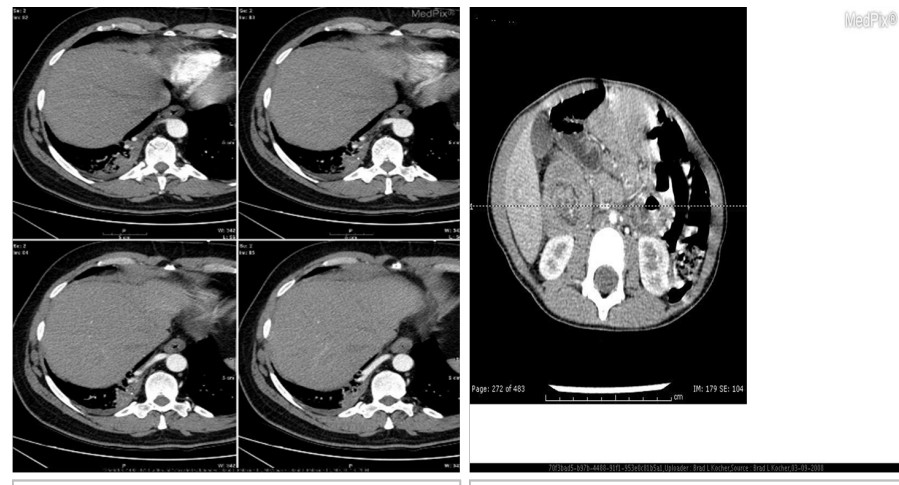

|  |  |
|---|---|
| Ground-truth: Selected axial CT images of the chest with IV contrast demonstrate an abnormal feeding vessel originating from the thoracic aorta coursing through an area of consolidation in the posterior segment of the right lower lobe. | Ground-truth: Axial CT images demonstrate a rounded mass in the right upper quadrant. |
| Generated: ct shows a lobe filling defect in right lower lobe and pulmonary lobe consistent of the lower lobe | Generated: axial CT images demonstrate a rounded mass in the right upper aspect of the right upper aspect of the right upper aspect of the right upper aspect of the right aspect of the mass |

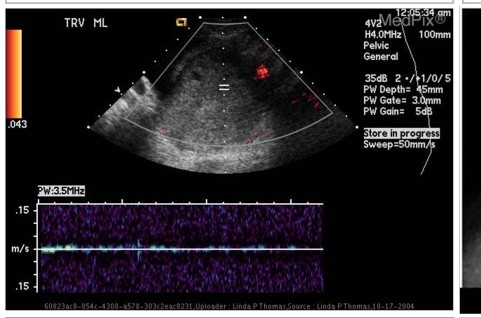 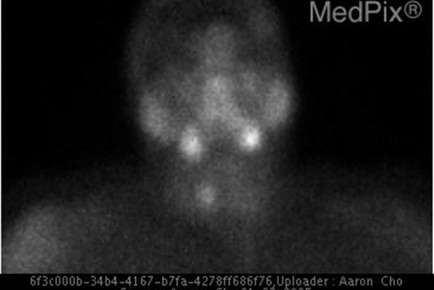

|  |  |
|---|---|
| Ground-truth: Gray scale and color doppler images reveal a large predominantly hyperechoic midline pelvic mass with multiple peripherally located cysts/follicles. Color Doppler arterial and venous waveforms absent. | Ground-truth: Tc99m Sestamibi scan demonstrates persistent tracer activity on 2hr delayed imaging consistent with parathyorid adenoma located at the level of the midpole of the right thyroid lobe. |
| Generated: transverse gray image of the mass with marked of the left pole to the left ovary | Generated: delayed image of the right mdp demonstrates area of tracer activity on the right femoral neck |

**Figure 13.** Samples of wrong captions generated by our system from images of bad quality. We illustrate in red the words that were wrongly inserted or missed by the model.

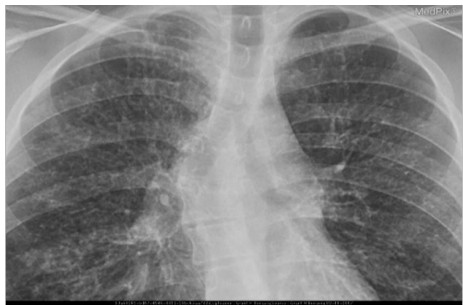

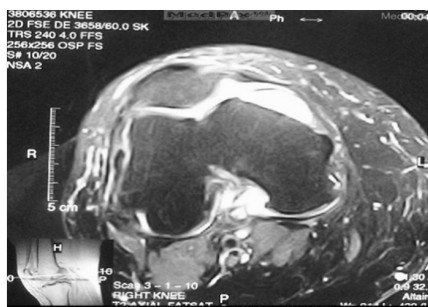

Ground-truth: Magnification of frontal radiograph of the chest demonstrating parahilar linear densities with peribronchial cuffing.

Generated: pa radiograph of the chest reveals multiple pleural opacity in the left upper venous in the left lung

Ground-truth: T2 Axial demonstrates hole in trochlear groove articular cartilage consistent with chronic patellar dislocation. Note severe effusion.

Generated: bony masses and lesion hematoma and metastases surrounding both superior and level associated spinal cord with hematoma destructive cm lesion in both

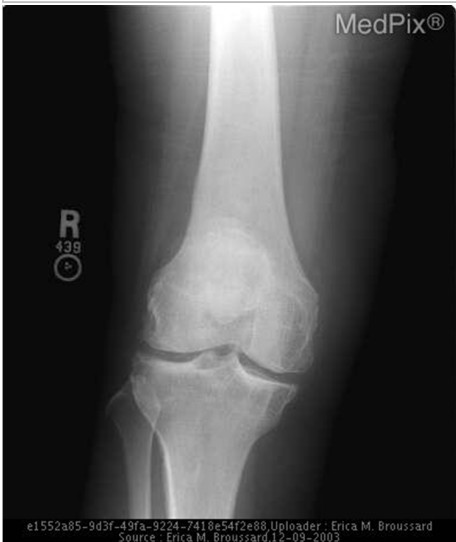

Ground-truth: AP view of the right knee demonstrates tricompartment joint space narrowing which is severe in the patellofemoral compartments. There is horizantally oriented calcification of the meniscal chondral cartilage.

Generated: frontal and lateral radiographs of the knee demonstrates sclerosis and metaphysis consistent with aspect of surface

Ground-truth: Fat-saturated T2-weighted coronal image reveals hyperintense intra- and extramuscular mass present within the forefoot. There are foci of T2 hyperintensity within the third and fourth metatarsals.

Generated: left saturated mr image of the left mass with enhancement in the region of the

**Figure 14.** Samples of wrong captions generated by our system from captions containing rare words. We illustrate in red the words that were wrongly inserted or missed by the model.

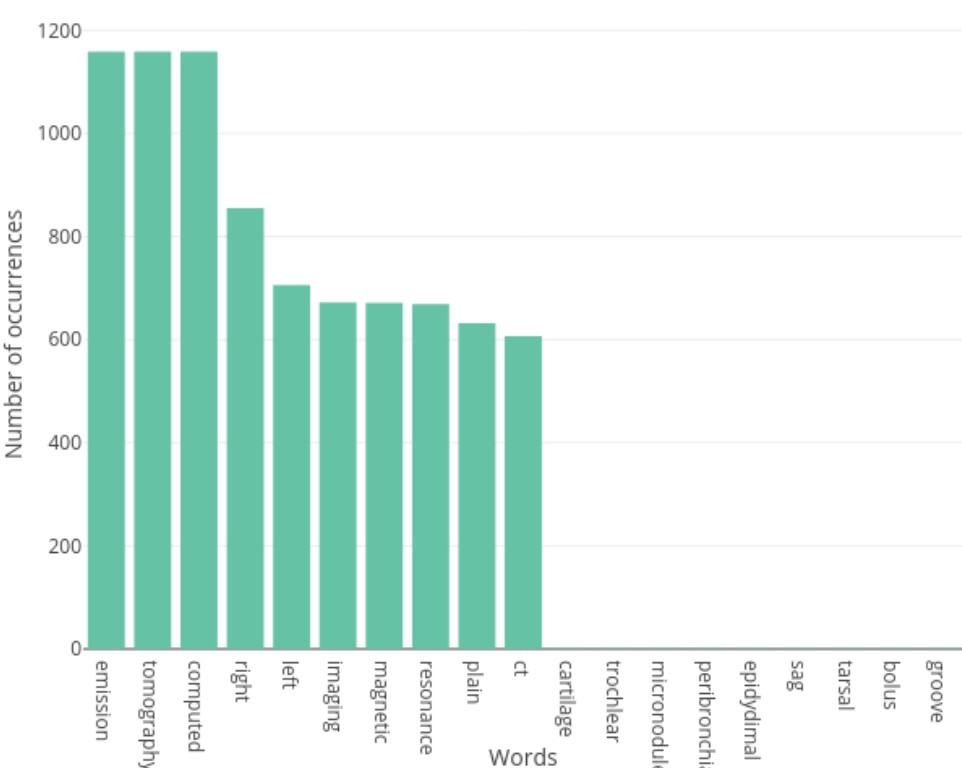

**Figure 15.** Some examples of words and their number of occurrences in the training captions. We show here the most frequent and rare words.

Finally, we illustrate in Figure 17 two samples of the generated captions for images where training samples pertain to different modalities (x-ray computed tomography and mammography modalities). In general, the model is shown as not being able to generate captions for images from modalities on which it has not been well-trained on. For instance, although the model predicted 'spiculated mass, the breast' for the left sub-figure, the model was not able to create a correct caption. Again, the model failed in determining the modality of the right sub-figure, where it was wrongly predicted as being an MRI image instead of an x-ray computed tomography.

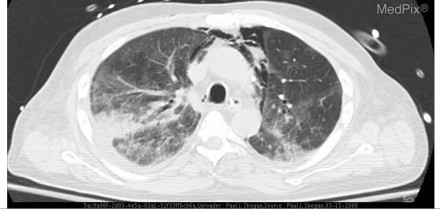

Ground-truth: On hospital day 10 the patient experienced worsening respiratory distress and an unenhanced CT of the chest was performed. An axial image at the level of the aortic arch demonstrates continued pneumomediastinum and a left-sided pneumothorax.

Generated: axial CT at the chest demonstrates multiple thickening multiple of the lung parenchyma

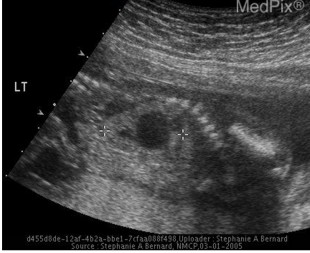

Ground-truth: There is severe hydronephrosis noted of the left kidney on axial image. On sagital imaging of the left kidney this is seen to primarily only involve the upper portion of the kidney. Mild pelviectasis is noted on the right within normal limits ( <4mm ).

Generated: sagittal sonographic image of the bladder

**Figure 16.** Samples of wrong captions generated by our system from captions containing numerical data or punctuation and characters that were clinically important. We illustrate in red the words that were wrongly inserted or missed by the model.

We show in Figure 18 the number of images per modality and the corresponding medical concept (modality). We can see that the training set was dominated by images from the tomography emission-computed modality, while few images were from the mammography modality. This justifies the number of occurrences of words 'tomography', 'computed', and 'emission' in the vocabulary of our model.

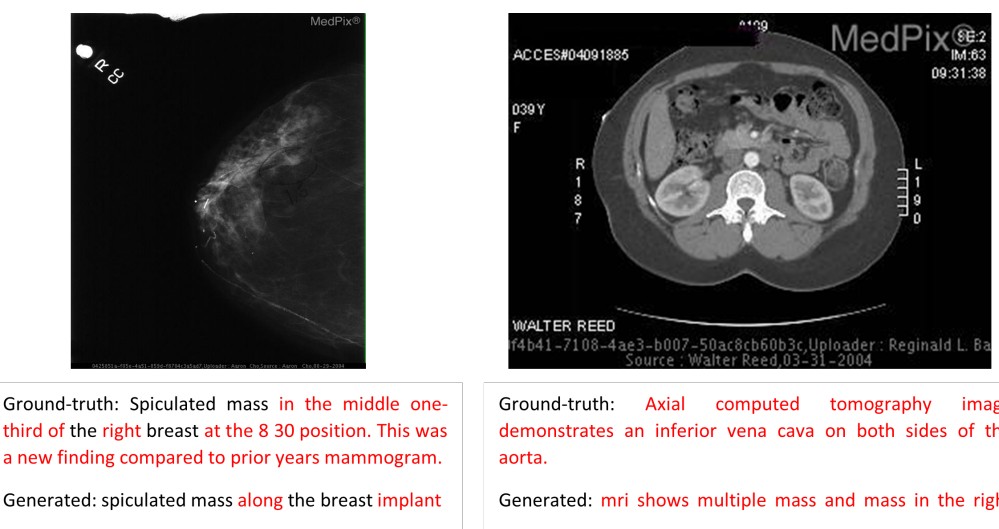

Ground-truth: Spiculated mass in the middle one-third of the right breast at the 8 30 position. This was a new finding compared to prior years mammogram.

Generated: spiculated mass along the breast implant

Ground-truth: Axial computed tomography image demonstrates an inferior vena cava on both sides of the aorta.

Generated: mri shows multiple mass and mass in the right

**Figure 17.** Samples of wrong captions generated by our system for images from modalities with few training samples. We illustrate in red the words that were wrongly inserted or missed by the model.

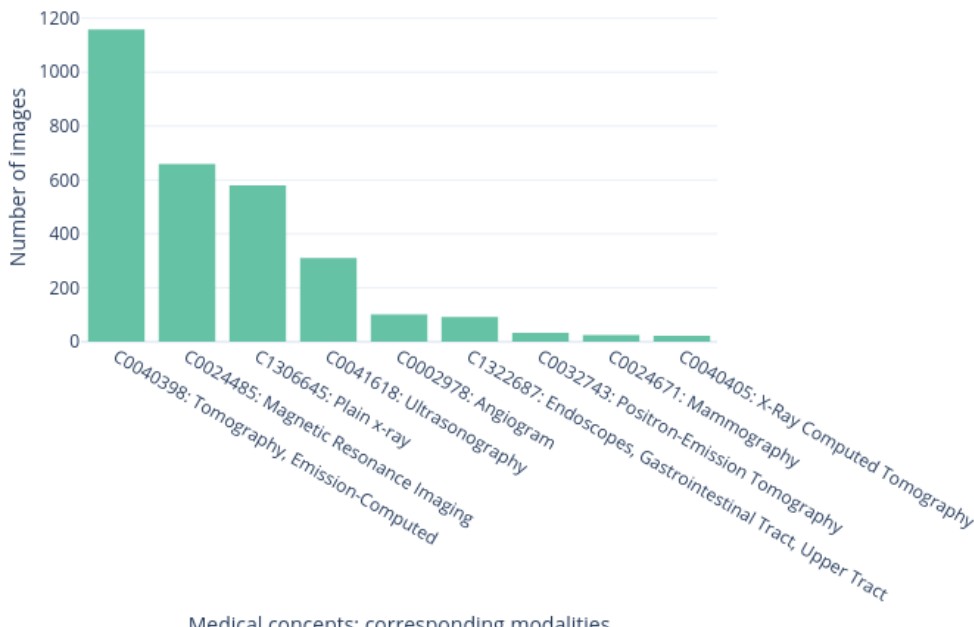

**Figure 18.** Number of images per modality and its corresponding medical concept.

To improve the results, it is possible to train the model more and fine-tune the hyperparameters to enhance accuracy performance. Some data augmentation techniques could be employed to resolve the issue of small training sample numbers. Indeed, data augmentation can be performed on both medical concepts and image captions to obtain more text samples. Different pre-trained embedding models could also be investigated, especially the ones trained on a large-scale medical corpus to cover the diverse terminology encountered in the training data, which can ultimately (potentially) improve the quality of the generated captions. Another direction would be to categorize the medical concepts

according to some topics and then perform a hierarchical multi-label classification (i.e., in two levels where the first level classifies the topics and the second one classifies the medical concepts). Then the predictions are made based on the two levels to better select the associated concepts of a given image. Finally, the pre-processing process should be performed more selectively, to allow us to keep clinically important data, even if they are numerical data or specific characters/symbols.

## 6. Conclusions

We presented an end-to-end network that integrates medical concepts alongside visual features extracted from original raw images for medical image captioning. The model is composed of different sub-models (a semantic feature encoder and visual feature encoder) trained simultaneously for a better generation of captions. The outputs of a visual encoder were merged with the outputs of a semantic encoder and inputted into the decoder to create captions. The visual encoder is a pre-trained network, the semantic encoder is a multi-label classifier, and the decoder is an LSTM network. A beam search with different indices was employed at the end to allow the LSTM to select better words for the newly created caption. Our model was evaluated on the ImageCLEF 2021 dataset by considering the images, their associated concepts, and their ground-truth captions as inputs of the model and generated captions as outputs. The experimental results and the ablation study illustrate the reliability of our method even though the quality of the generated captions still needs to be improved. For that, more training, hyperparameter optimization, and pre-processing have to be taken into account to improve the findings.

**Author Contributions:** Conceptualization, D.R.B. and M.O.; methodology, D.R.B. and M.O.; software, D.R.B.; validation, D.R.B. and M.O. and R.J.; formal analysis, D.R.B. and M.O.; investigation, D.R.B.; resources, D.R.B. and M.O.; data curation, D.R.B.; writing—original draft preparation, D.R.B.; writing—review and editing, D.R.B. and M.O.; visualization, D.R.B.; supervision, M.O. and T.S. and R.J.; funding acquisition, T.S. All authors have read and agreed to the published version of the manuscript.

**Funding:** This work is supported by the Academy of Finland Profi5 DigiHealth project (#326291), which is gratefully acknowledged.

**Institutional Review Board Statement:** Not applicable.

**Informed Consent Statement:** Not applicable.

**Data Availability Statement:** Not applicable.

**Conflicts of Interest:** The authors declare no conflict of interest. The funders had no role in the design of the study; in the collection, analyses, or interpretation of data; in the writing of the manuscript; or in the decision to publish the results.

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
