# Peer review of "ACapMed: Automatic Captioning for Medical Imaging"

_applsci, doi:10.3390/app122111092_

Round 1

Reviewer 1 Report

In this article, the authors combined some deep learning models of image processing and natural language processing models to design a method to automatically generate captions illustrating the diagnosis for medical images. The goal of this article fitted the direction of the automated medical image interpretation and the scope of the special issues. The methods were clearly described. The design of the deep learning model was reasonable in terms of the properties of the medical images and the purpose to generate captions. The performance of the model was comparable to other model for the similar purposes. The referee recommend publication after minor changes.

The authors demonstrated some prediction examples in validation and described whether the model successfully generate the consistent captions or not. However, it lacks error analysis to explain why the model failed in some of the cases. It is suggested the authors manually review a. certain amount of fail cases and sort them into different categories according to the possible reasons of misinterpretation (e.g. not enough images for a certain type, image quality issues, flaws in the original caption, edge cases with words rarely used,….). The error analysis can be combined with the discussion section for the future directions of model improvement, as the improvement should aim at dealing with these errors.

Minor typo: Line 77: VGG1-16 should be corrected too VGG-16

Reviewer 2 Report

Based on a careful analysis, I can formulate the following remarks:

1)  The aim of this article, based on the author’s scrupulous investigations, is to solve the problem of the efficient information extraction from the captured medical images. They offer an end-to-end network that integrates medical concepts alongside visual features extracted from original raw image for medical image captioning. The model is composed of different sub-models (semantic feature encoder and visual feature encoder) trained simultaneously for a better generation of captions.

2) The topic represents in my opinion a relevant approach of the proposed theme, based on meticulous theoretical and investigations, correlated with relevant experimental results.

The outputs of a visual encoder, merged with the outputs of a semantic encoder and inputted to the decoder assure the desired captions’ creation. The visual encoder is a pre-trained network while the semantic encoder is a multi-label classifier and the decoder is a Long Short Term Memory network (LSTM) network. Beam search with different indices is employed at the end to allow the LSTM to select better words of the newly created caption.

3) In comparison with other published material, the author’s contribution adds to the subject area a new approach/methodology, with the following significant contributions:

• A new approach to generate Vocabulary for text generation and encoding is constructed from medical concepts and image captions;

• A multi-label classification (MLC) model based on the VGG-16 type model was involved to detect medical concepts from images. This VGG-16 model, having 16 layers, is very suitable for the proposed technique.

• A pre-trained VGG1-16 network is employed to extract visual features from medical images.

• An end-to-end deep learning based network is employed for text generation fusing both visual and semantic features extracted from images as well as their associated medical concepts.

The model merges different networks: MLC for semantic features encoding, VGG-16 for visual features extraction, captions embedding and an LSTM network for caption generation.

• A beam search is employed alongside LSTM to accurately select the best word among the list of predicted words for caption construction.

• An ablation study is conducted to investigate the impact of each component of the proposed model in the overall performance.

The proposed technique, evaluated on a publicly available medical dataset, demonstrate the feasibility and the technical soundness of the authors’ method, better than the other reported ones from the literature.

4) The issue of the medical image captioning is a very challenging task, which has been less in the literature of natural image captioning. Some existing image captioning techniques exploit objects present in the image side by side to the visual features while generating descriptions.

However, this is not possible for medical image captioning when one requires following clinician-like explanation in image content description.

Based on this, the authors propos to use medical concepts associated to images in line with their visual features to generate new caption. Their end-to-end trainable network is composed of a semantic feature encoder based on a multi-label classifier to identify medical concepts related to images, a visual feature encoder, and an LSTM model for text generation.

The beam search ensures a best selection of the next word for a given sequence of words based on the merged features of the medical image.

They evaluated their proposal on the ImageCLEF medical captioning dataset, and the results demonstrate the effectiveness and efficiency of the developed approach.

5) In my opinion, the presented conclusions are suitable related to their research results and prove that they reached the proposed goal.

6) The references in my opinion are very appropriate and their number underlines the scrupulosity of the authors.

7) Notably, classical captioning models struggle to generate accurate descriptions for medical images and still need improvements to be clinically acceptable.

Therefore, implementation of new approaches tailored for medical image captioning is an emerging field of great interest in artificial intelligence. This may help in fast exploitation of medical content, delivery of faster and more accurate interpretations of the findings, which provide valuable assistance to doctors by alleviating their workload and expediting the clinical workflows.

Although major effort has been made to enhance the quality of natural image captioning boosted by commercial and security related application, little advance is made in

medical image captioning field where the accuracy barely exceeds 40% in several bench-marking competitions.

In this regard, the authors offer in this contribution an efficient extraction of the medical concepts from the image content and employ them inline with its visual features to generate a new caption. The motivation behind this work comes from the use of object detection algorithms in image captioning systems.

In order to validate the proposed original approach, the author underlined the significance of testing using the real datasets. Based on the validation results, one can underline the efficiency of their proposed approach.

 In conclusion: a very promising approach, which can be and has to be continued.

One has to remark the fact that both the graphical illustrations as well as the figures are very suggestive.

I encourage publishing in a new contribution his further results.
